# Simulation of Temperature Field in Micro-EDM Assisted Machining of Micro-Holes in Printed Circuit Boards

**DOI:** 10.3390/mi13050776

**Published:** 2022-05-15

**Authors:** Manqun Lian, Xinke Feng, Bin Xu, Lianyu Fu, Kai Jiang

**Affiliations:** 1Shenzhen Key Laboratory of High Performance Nontraditional Manufacturing, Shenzhen University, Shenzhen 518060, China; lianmanqun2020@email.szu.edu.cn (M.L.); fengxinke2019@email.szu.edu.cn (X.F.); binxu@szu.edu.cn (B.X.); 2Shenzhen Jinzhou Precision Technology Corp., Shenzhen 518116, China; mlyfu@chinadrill.com; 3College of Software Engineering, Shenzhen Institute of Information Technology, Shenzhen 518172, China

**Keywords:** micro-EDM, printed circuit board (PCB) micro-hole, temperature field, micro-bit wear

## Abstract

High-speed mechanical drilling based on the micro-bit is the mainstream process technology for machining micro-holes in the printed circuit board (PCB). However, the above process to obtain PCB micro-holes is prone to defects, such as hole burrs and nail heads in the hole. In this paper, the micro electrical discharge machining (micro-EDM) was used as an auxiliary means for machining PCB micro-holes to effectively eliminate the defects such as hole burrs and nail heads. However, during the process of micro-EDM, the micro-bit will be gradually worn, thus negatively affecting the machining quality of PCB micro-holes. To solve the above problems, in this paper, the temperature field model of micro-EDM-assisted machining of PCB micro-holes was established to predict the micro-bit wear by analyzing the temperature field with COMSOL Multiphysics software. This paper made an extensive study of the influences of spindle speed, machining voltage, and pulse width on temperature field and micro-bit wear. The simulation results show that with the increase in machining voltage and pulse width, the temperature of PCB micro-hole machining increases, resulting in an increase in micro-bit wear. The spindle rotation is beneficial to the updating of the machining medium and the discharge of heat generated from EDM. Therefore, with the increase in spindle speed, the temperature of PCB micro-hole machining and the micro-bit wear is reduced.

## 1. Introduction

The printed circuit board (PCB) micro-holes provide positioning and support for electronic components and can transmit signals. High-speed mechanical drilling based on micro-bit is the mainstream process technology of PCB micro-hole machining. However, PCB micro-holes obtained by mechanical drilling are susceptible to defects, such as hole burrs and nail heads. Hole burrs may cause a short circuit of PCB, and nail heads in the hole may lead to conductive failure during the preparation of the plated hole. Micro electrical discharge machining (Micro-EDM) is a non-contact machining technology, which has the advantages of low cutting force and high machining precision [1]. At present, micro-EDM is used as an auxiliary means for the machining of copper foil layer in PCB, to effectively avoid the defects such as hole burrs and nail heads in PCB micro-hole [2].

To improve the EDM efficiency of Inconel 718, Tanjilul et al. developed a new super dielectric fluid [3]. Compared with conventional dielectrics, the fluid can effectively improve EDM efficiency. To improve the EDM efficiency and machining quality of micro-holes, Shin et al. developed a TR pulse generator [4]. Through EDM, Singh et al. processed micro-holes in Ti-6Al-4V alloy material [5] and the results showed that the removal of electric corrosion products and the renewal of the machining medium had great effects on the machining quality of micro-holes. Kumar et al. conducted EDM experiments on Ti-6Al-7Nb to machine micro-holes [6]. The results showed that the cylindrical electrode was depleted into a cone shape, which adversely affected the shape accuracy of the micro-holes. To improve the EDM accuracy of micro-holes, Liang et al. proposed a self-repairing method and a contour error compensation method for electrode wear [7]. Tang et al. established a multi-physics model including thermal, hydraulic, metallographic, and structural mechanics to research the thermal phase transformation and residual stress during single-pulse EDM of Ti–6Al–4V titanium alloys [8]. To research the effect of alternating current on micro-EDM, Yang et al. established a discharge model based on real experimental waveforms [9]. Choubey et al. developed a finite element simulation model for dimples and material removal rate (MRR) in micro-electro-discharge machining (micro-EDM) with and without vibration of the workpiece [10]. To research wire-cut electrical discharge machining (wire-cutting) of polymer composites (PCM), Ablyaz et al. established a theoretical model to analyze this process. This model was used to analyze the heat flux poles available on micro-EDM electrodes and workpieces [11]. Li et al. adopted the simulation of the ultrasonic circular vibration (UCV) machining process with the solid–liquid two-phase flow and studied the flow of field distribution and discharge debris movement in the machining gap [12].

In order to obtain the micro-EDM wear of the electrode, Heo et al. developed a virtual EDM simulator to perform the evolution process of electrode shape [13]. Somashekhar et al. made a simulation study of the machining phenomenon of micro-EDM by the finite element method [14] and obtained the temperature field distribution. Quarto et al. developed simulation software for micro-EDM by the finite element method and used the software to simulate the EDM process of micro-holes [15]. Nadda et al. proposed a mathematical model based on the electrothermal principle to analyze the size and morphology of micro-pits produced by EDM [16]. Yue et al. adopted a two-temperature model (TTM) to simulate the EDM machining process [17] and the simulation results were close to the experimental results.

In the micro-EDM-assisted machining of PCB micro-holes, the micro-bit will gradually wear, thus negatively affecting the machining quality of PCB micro-holes. Through the simulation and calculation of the above process [18], the temperature field distribution law and the prediction of micro-bit wear can effectively improve the PCB micro-hole machining quality. In this paper, the temperature field model of micro-EDM-assisted machining of PCB micro-holes was established, and the temperature field was analyzed by COMSOL Multiphysics software so as to predict the micro-bit wear.

## 2. Simulation Model

PCB is composed of copper foil, modified resin, ceramic filler, and glass fiber cloth. With the help of an electric spindle and motion platform, the micro-bit with high-speed rotation completed the drilling of modified resin, ceramic filler, and glass fiber. With a high-frequency pulse power supply, electric spark discharge was generated between the micro-bit and PCB copper foil layer, and copper foil was machined in the form of debris through electric erosion, thus effectively avoiding the generation of hole burrs and nail heads in the hole. In this paper, the temperature field of micro-EDM-assisted machining of PCB micro-holes was simulated. 

### 2.1. Assumptions of Simulation Conditions

During the micro-EDM, the electric field, temperature field, and magnetic field are coupled with each other, thus making the process of electric spark discharge very complicated. Due to the above factors, it is very difficult to realize the precise simulation of the EDM process. Therefore, in this paper, the simulation model will be simplified. Based on the actual conditions, the following assumptions are made for the simulation model:

(1) A single pulse produces only one discharge channel with a regular cylinder shape. The voltage and current in the discharge channel remain stable during the process of discharge pulse action.

(2) At the moment of discharge generation, the discharge channel will not deform with the rotation of the micro-bit.

(3) The micro-bit is made of isotropic materials. Micro-bit transfers heat by means of heat conduction.

(4) Heat transfer between the micro-bit and the spark oil is conducted by convection, and the convective heat transfer coefficient remains stable during the discharge process.

(5) The heat-affected zone produced by single-pulse discharge is infinitely small relative to the micro-bit.

### 2.2. Establishment of Heat Transfer Model

Based on the above assumptions, a stable cylindrical discharge channel will be generated between the micro-bit and the copper foil during spark discharge. The schematic diagram of the discharge channel is shown in Figure 1.

The energy (*W_t_*) produced by the discharge can be calculated from the following equation:(1)$$W_{t}=\int_{0}^{t_{on}}U_{d}I_{d}dt$$

*U_d_* is the machining voltage, *I_d_* is the discharge current, *t_on_* is the pulse width. The energy (*W_t_*) generated by the discharge is distributed to the cathode, anode, and processing medium in a specific proportion. The energy (*W_e_*) distributed to the cathode can be calculated from the following equation:(2)$$W_{e}=W_{t}F_{e}$$

*F_e_* is the cathode energy distribution coefficient. The average heat flux (*H_e_*) in the discharge channel of the cathode surface can be calculated from the following equation:(3)$$H_{e}=\frac{U_{d}I_{d}F_{e}}{\pi R_{d}{}^{2}}$$

*R_d_* is the radius of the discharge channel. The heat flux density in the discharge channel is not uniformly distributed, and the heat flux density is the highest in the center of the discharge channel. Taking the center of the discharge channel as the basic point, the heat flux shows a normal distribution. Therefore, this paper will be based on the Gaussian heat source. The heat flux density (*H_d_*) on the micro-bit surface within the discharge channel can be calculated from the following equation:(4)$$H_{d}=4.45H_{e}exp\left(-\alpha \frac{R^{2}}{R_{d}{}^{2}}\right)$$

*R* is the distance of any point from the discharge center, *α* is the adjustment coefficient of discharge concentration degree, taking the value of 4.6. In addition to the heat generated by the electric spark discharge, there is convective heat transfer between the micro-bit surface and the spark oil. The heat flux density (*H_c_*) can be calculated from the following equation:(5)$$H_{c}=h\left(T-T_{0}\right)$$

*h* is the convective heat transfer coefficient, taking the value of 800 Wm^−2^K^−1^. *T* is the surface temperature of the micro-bit and *T*_0_ is the spark oil temperature. Based on the above analysis, the overall heat flux density (*H_T_*) of the micro-bit surface can be calculated from the following equation:(6)$$H_{T}=\{\begin{array}{c} 4.45\frac{U_{d}I_{d}F_{e}}{\pi R_{d}{}^{2}}exp\left(-4.6\frac{R^{2}}{R_{d}{}^{2}}\right)+h\left(T-T_{0}\right),\;\;\left(R\leq R_{d}\right)\\ h\left(T-T_{0}\right),\;\;(R>R_{d}) \end{array}$$

The overall heat flux distribution on the micro-bit surface is shown in Figure 2.

### 2.3. Simulation Conditions

The modeling of the micro-bit is shown in Figure 3. A micro-element of 40 μm ∗ 20 μm ∗ 10 μm on the surface of the micro-bit is taken as the object of finite element analysis. The discharge occurs on the upper surface of the micro-element, that is A plane. The XYZ coordinate system is established with the discharge center on A plane as the origin. The boundary conditions are as follows:

(1) Heat flux of the A plane is *H_T_*.

(2) All surfaces other than A plane are thermally insulated.

(3) The initial temperature of both the micro-element and the environment is 25 °C.

In Figure 3, it can be seen that the physical field is symmetrical with respect to the XZ plane during the discharge. The model is simplified to reduce the computational amount of simulation and improve computation efficiency. As shown in Figure 4, the simplified finite element model of micro-bit defines the symmetric boundary condition on B_1_ plane. The above simplification of finite element modeling can reduce the computational amount by half without affecting the simulation accuracy.

As shown in Figure 1, the handle of the micro-bit is made of high-speed steel (HSS) and the edge material is made of YG8 cemented carbide. Based on the material library of COMSOL Multiphysics software system, the properties of YG8 material relevant to this paper are shown in Table 1.

The finite element division is carried out on the micro-element of the micro-bit, and the results are shown in Figure 5. In this paper, in order to ensure the computational accuracy of the temperature field, the mesh division during the discharge is refined, and the mesh length after the refinement process is 0.14 μm.

During the discharge, along with the rotation of the micro-bit, the center of the heat source moves along the X axis in a straight line from the coordinate origin with a uniform moving speed (*v*). The distance (*R*) from the discharge center can be calculated by the Equation (7):(7)$$R=\sqrt{{\left(x-vt\right)}^{2}+y^{2}}$$

The cathode energy distribution coefficient (*F_e_*) can be calculated from the Equation (8) [19]:(8)$$F_{e}=8.15I_{d}{}^{-0.49}t_{on}{}^{0.62}$$

The discharge channel radius (*R_d_*) can be calculated from the Equation (9) [20]:(9)$$R_{d}=2.04I_{d}{}^{0.43}t_{on}{}^{0.44}$$

*R* is the distance of any point from the discharge center, *F_e_* is the cathode energy distribution coefficient, *R_d_* is the discharge channel radius, *I_d_* is the discharge current, and *t_on_* is the pulse width. The existing research results show that the quality of micropores processed under the action of 10–80 V voltage is relatively better [21]. In order to determine the value of discharge current, a series of PCB micro-hole machining experiments were carried out with the machining voltages ranging from 10 V to 81 V. Additionally, an oscilloscope is used to measure the peak current of the spark discharge, and the results are shown in Table 2.

The relationship between the discharge current (*I_d_*) and the machining voltage (*U_d_*) can be obtained by linear fitting the data shown in Table 2.
(10)$$I_{d}=0.00938U_{d}-0.01173$$

Figure 6 shows the correspondence between the machining voltage and the discharge current. The results show that the trend presented by the Equation (10) coincides well with the actual machining data. Therefore, in this paper, the discharge current values of different machining voltages can be calculated from the Equation (10).

## 3. Results and Discussion

### 3.1. Simulation of the Temperature Field of the Micro-bit

According to the actual machining experiment, when the spindle speed was 20,000 rpm, the machining voltage was 25 V and the pulse width was 33 μs; thus, the machining quality of the PCB micro-hole was better. In this paper, the temperature field of the micro-bit was simulated with the above process parameters, and the simulation results are shown in Figure 7.

According to the simulation results in Figure 7, the temperature of the discharge center region is the highest, and the temperature of the region far away from the discharge center decreases sharply. In addition, the position of the highest temperature point gradually moves to the right as the EDM process proceeds. As shown in Figure 8, *T_max_* and *T*_(0,0,0)_ increase rapidly in a short period of time during the initial stage. *T_max_* increases slowly and stabilizes as discharge progressed. When the machining time is 3 μs, *T_max_* reached the maximum value of 8260 °C. As the discharge proceeds, *T*_(0,0,0)_ rises firstly and then decreases. When the machining time is 0.72 μs, *T*_(0,0,0)_ reaches the maximum value of 7747 °C. When the machining time exceeds 0.72 μs, *T*_(0,0,0)_ begins to decrease slowly.

Under the action of a high-frequency pulsed power supply, the temperature of the micro-bit continues to rise and the high-temperature area is concentrated in the area where the discharge channel is generated. In the high-temperature area of the micro-bit, heat is transferred to the low-temperature area by conduction and to the spark oil in the form of convection. The greater the temperature difference between the high-temperature area and the low-temperature area, the greater the effect of both heat transfers. Therefore, when the temperature of the discharge area is low, the heat dissipation is small, which leads to a rapid rise in *T_max_*. As the temperature of the discharge area increases, the heat loss caused by heat transfer increases. When the heat generated by the discharge is balanced with the heat loss, *T_max_* also stabilizes.

At the beginning of the discharge, the center of the discharge channel is at (0,0,0). Under the action of a high-frequency pulse power supply, *T*_(0,0,0)_ rises rapidly. With the progress of EDM, the center of the discharge channel gradually moves away from the starting point, and the heat generated by the discharge gradually weakens against the starting point. Under this condition, *T*_(0,0,0)_ begins to decrease gradually.

As shown in Figure 8, the maximum temperature of the micro-bit can reach 8260 °C. The melting point of YG8 cemented carbide is 2800 °C. Therefore, under the effect of high temperature generated by electric discharge, the micro-bit will melt and vaporize, resulting in the EDM wear. In this paper, the temperature field of the micro-bit over its melting point is noted as *V_m_*, and *V_m_* is equivalent to the EDM wear of the micro-bit (Figure 9).

### 3.2. Effects of Machining Voltages on the Temperature Field of the Micro-bit

In this paper, a series of simulation calculations were carried out using the machining voltages ranging from 15 V to 80 V to study the effects of machining voltages on the temperature field and the micro-bit wear. The pulse width was set to 3 μs and the spindle speed was set to 20,000 rpm. The variation of *T*_(0,0,0)_ and *T_max_* with different machining voltages is shown in Figure 10. The variation of the micro-bit wear volume (*V_m_*) with different machining voltages is shown in Figure 11. According to the simulation results, with the progress of EDM, *T*_(0,0,0)_ rises rapidly and then decreases slowly. *T_max_* rises rapidly and then stabilized. In addition, the larger the machining voltage is, the larger the *T*_(0,0,0)_ and *T_max_* are. When the voltage is 15 V, the maximum value of *T*_(0,0,0)_ is 4493 °C and that of *T_max_* is 4768 °C. When the voltage is 80 V, the maximum value of *T*_(0,0,0)_ is 26,797 °C and that of *T_max_* is 28,712 °C. As shown in Figure 11, the wear volume (*V_m_*) of the micro-bit gradually increases with the increase in the processing voltage. When the voltage is 15 V, *V_m_* is 7.6292 × 10^−19^ m^3^. When the voltage is 80 V, *V_m_* is 8.4188 × 10^−16^ m^3^.

In the initial stage of EDM, *T*_(0,0,0)_ and *T_max_* rise rapidly under the heat loss caused by both electric discharge and heat transfer. When the heat generated by the discharge is balanced with the heat loss, *T_max_* also stabilizes. With the progress of EDM, the center of the discharge channel gradually moves away from the starting point, so *T*_(0,0,0)_ decreases gradually with the increase in machining time. As the machining voltage increases, the energy generated by spark discharge increases, so *T*_(0,0,0)_ and *T_max_* increase gradually. Under the above conditions, the temperature of the micro-bit is larger than the melting point of YG8 cemented carbide, leading to the increase in the micro-bit wear volume (*V_m_*).

### 3.3. Effects of Pulse Widths on the Temperature Field of the Micro-bit

In order to obtain the effect of pulse width on the temperature field and micro-bit wear, a series of simulation calculations of the temperature field were carried out with the pulse widths from 1 μs to 10 μs. In the simulation calculation, the machining voltage was set to 25 V and the spindle speed was set to 20,000 rpm. The variation of *T*_(0,0,0)_ and *T_max_* with machining time for different pulse widths is shown in Figure 12. The micro-bit wear volume (*V_m_*) of the simulation is shown in Figure 13. As shown in Figure 12a, with the progress of EDM, *T*_(0,0,0)_ rises rapidly first and then decreases slowly. The larger the pulse width is, the larger the *T*_(0,0,0)_ is. When the pulse width is 1 μs, the maximum value of *T*_(0,0,0)_ is 6456 °C. When the pulse width is 10 μs, the maximum value of *T*_(0,0,0)_ is 9463 °C. As shown in Figure 12b, with the progress of EDM, *T_max_* rises rapidly and then stabilizes. The larger the pulse width is, the larger the *T_max_* is. When the pulse width is 1 μs, the maximum value of *T_max_* is 6372 °C. When the pulse width is 10 μs, the maximum value of *T_max_* is 11,645 °C. As shown in Figure 13, the micro-bit wear volume (*V_m_*) gradually increases as the pulse width increases. When the pulse width is 1 μs, *V_m_* is 4.1895 × 10^−19^ m^3^. When the pulse width is 10 μs, *V_m_* is 1.3642 × 10^−17^ m^3^.

The Equation (1) shows that as the pulse width (*t_on_*) increases, the energy (*W_t_*) generated by the discharge increases, and the cathode energy distribution coefficient (*F_e_*) increases. According to the Equation (2), as the pulse width (*t_on_*) increases, the energy (*W_e_*) distributed to the cathode increases. Due to the above factors, the temperature of the micro-bit increases, and causes a gradual increase in *T*_(0,0,0)_ and *T_max_*. When the heat generated by the discharge is balanced with the heat loss, *T_max_* increases slowly with the progress of machining. With the progress of EDM, the center of the discharge channel gradually moves away from the starting point. Meanwhile, the effect of the energy generated by the discharge on the starting point becomes smaller and smaller, so *T*_(0,0,0)_ decreases gradually with the increase in machining time. The above analysis shows that as the pulse width (*t_on_*) increases, the energy (*W_e_*) distributed to the cathode increases, and the temperature in the discharge region of the micro-bit increases. Due to the above factors, the micro-bit wear volume (*V_m_*) increases.

### 3.4. Effects of Spindle Speed on the Temperature Field of the Micro-bit

In order to investigate the effect of spindle speed on the temperature field and micro-bit wear, the simulation calculations were carried out with the spindle speeds from 10,000 rpm to 80,000 rpm. In the simulation calculation, the machining voltage was set to 25 V and the pulse width was set to 3 μs. The variation of *T*_(0,0,0)_ and *T_max_* with machining time for different spindle speeds is shown in Figure 14. The micro-bit wear volume (*V_m_*) in the simulation is shown in Figure 15. As shown in Figure 14a, with the progress of EDM, *T*_(0,0,0)_ rises rapidly first and then slowly decreases. In addition, the larger the spindle speed is, the faster *T*_(0,0,0)_ decreases. When the spindle speed is 10,000 rpm, the maximum value of *T*_(0,0,0)_ is 7912 °C. When the spindle speed is 80,000 rpm, the maximum value of *T*_(0,0,0)_ is 7282 ℃. As shown in Figure 14b, the variation trend of *T_max_* is consistent under different spindle speeds. At the initial stage of EDM, *T_max_* rises rapidly and then stabilizes. As shown in Figure 15, with the increase in spindle speed, the micro-bit volume (*V_m_*) increases firstly and then decreases. When the spindle speed is 10,000 rpm, *V_m_* is 1.0527 × 10^−17^ m^3^. When the spindle speed is 30,000 rpm, *V_m_* is 1.0530 × 10^−17^ m^3^, reaching its maximum. When the spindle speed is 80,000 rpm, *V_m_* is 1.0453 × 10^−17^ m^3^.

In the initial phase of EDM, the heat generated by spark discharge causes *T*_(0,0,0)_ and *T_max_* to rise rapidly in a short period of time. The rotational movement of the spindle facilitates the removal of heat from the discharge area. The faster the spindle speed, the faster the heat is dissipated from the discharge area. As a result, *T*_(0,0,0)_ gradually starts to fall under the effect of heat dissipation. The faster the spindle speed, the faster *T*_(0,0,0)_ falls. As the spark discharge occurs in a very short time, the spindle speed does not have enough time to influence the discharge energy. Therefore, the spindle speed hardly affects the energy input at the highest temperature of the discharge area. Due to the above factors, the variation trend of *T_max_* under the effect of different spindle speeds remains the same.

As the spindle speed increases, the area sweeps by the discharge channel on the micro-bit surface increases per unit time, thus allowing more micro-bit area to generate spark discharge. As a result of the discharge energy, the temperature of the spark discharge area rises. When the temperature exceeds the melting point of the micro-bit, the micro-bit will be worn by EDM. Therefore, when the spindle speed is between 10,000 rpm and 30,000 rpm, *V_m_* increases as the spindle speed increases. When the spindle speed exceeds 30,000 rpm, although the discharge channel sweeps over a larger area of the micro-bit surface per unit time, the discharge heat input time at each point on the discharge channel path is too short. Under this condition, the area where the temperature is higher than the melting point of the micro-bit becomes smaller, thus *V_m_* decreases.

## 4. Model Verification

### 4.1. Experimental Design

In this paper, the three-axis micro-EDM-assisted drilling machine for PCB micro-holes with a gantry structure is based on the precision displacement platform (M155, PI), as shown in Figure 16. The maximum stroke of the experimental platform is 102 mm × 102 mm × 102 mm, the maximum feed rate is 100 mm/s, and the positioning accuracy is 50 nm. The spindle (BM-320F, NAKANISHI) used in this experimental platform has a rotational speed adjustment range of 1000 rpm to 80,000 rpm, and the spindle runout accuracy is within 1 μm. The high-frequency pulse power supply independently developed by the laboratory is used to power the micro-EDM, the voltage adjustment range is 0 V to 200 V, and the minimum pulse width is 1 μs. During processing, the PCB is immersed in EDM special kerosene (EDM-XTRA, Mobil).

### 4.2. Experimental Results

In order to verify the simulation results, experiments of micro-EDM-assisted machining of PCB micro-holes were carried out with different machining voltages, pulse widths, and spindle speeds. The morphology of the micro-bit was observed by SEM, and the experimental results are shown in Figure 17.

As shown in Figure 17a, the EDM wear of the micro-bit gradually increases as the machining voltage increases. As shown in Figure 17b, the EDM wear of the micro-bit gradually increases as the pulse width increases. As shown in Figure 17c, when the spindle speed is 30,000 rpm, the wear of the micro-bit is maximum. When the spindle speed exceeds 30,000 rpm, the wear of the micro-bit decreases gradually. The wear trend of the micro-bit presented by the above experimental results is basically consistent with the trend presented by the numerical simulation results, thus verifying the feasibility of the simulation model established in this paper.

## 5. Conclusions

The micro-bit wear can affect the machining quality of PCB micro-holes. In this paper, the temperature field model of micro-EDM-assisted machining of PCB micro-holes was developed, and the temperature field was analyzed by COMSOL Multiphysics software so as to predict the EDM wear of the micro-bit. The main conclusions of this paper are described as follows.

(1) This paper used COMSOL Multiphysics software to build a simulation model of the temperature field of micro-EDM-assisted machining of PCB micro-holes. During the discharge process, the heat generated by the discharge will make the temperature of the micro-bit rise rapidly at the beginning of the discharge. With the progress of the discharge, the temperature at the starting point of the discharge center will decrease gradually, while the maximum temperature inside the micro-bit will continue to rise slowly and then stabilize, reaching a state of dynamic equilibrium. The micro-bit will melt and vaporize under the high temperature generated by the electrical discharge. 

(2) In this paper, the effects of machining voltage, discharge pulse width, and spindle speed on the wear of the micro-bit were analyzed. With the increase in machining voltage and discharge pulse width, the energy of a single pulse discharge will increase, which will result in larger heat input to the surface of the micro-bit, thus increasing the temperature and the lost volume of the micro-bit. The increase in spindle speed will make the discharge channel move faster across the micro-bit surface, causing the wear volume of the micro-bit to increase first and then decrease. Therefore, in order to reduce the wear of the micro-bit, the machining voltage and discharge pulse width can be properly reduced and the spindle speed can be increased while ensuring the machining quality.

## Figures and Tables

**Figure 1 micromachines-13-00776-f001:**
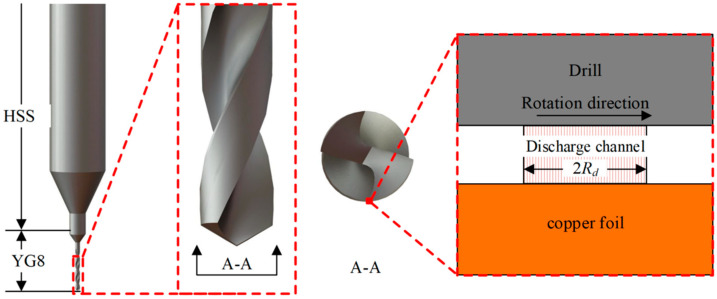
Discharge channel.

**Figure 2 micromachines-13-00776-f002:**
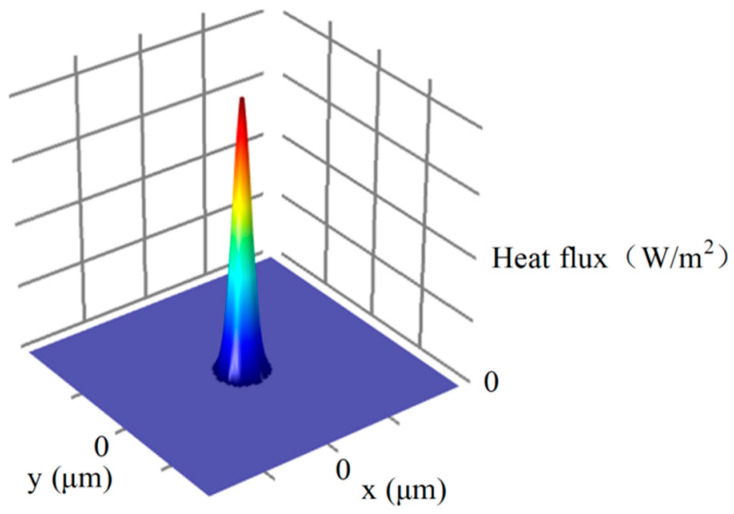
Heat flux density of the micro-bit surface.

**Figure 3 micromachines-13-00776-f003:**
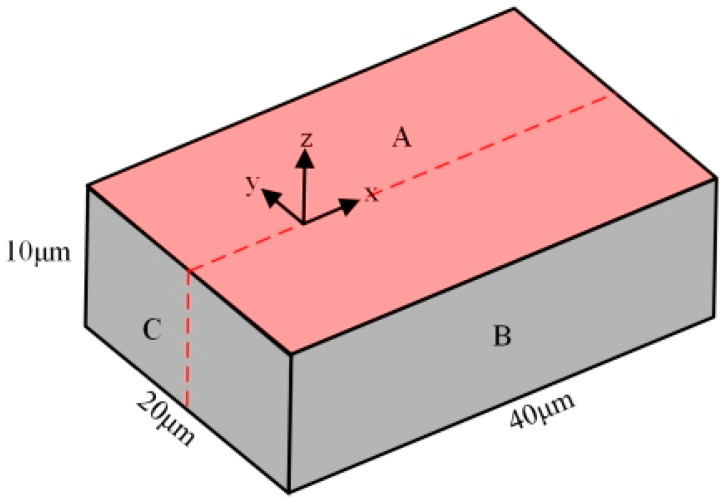
Micro-element on the micro-bit surface.

**Figure 4 micromachines-13-00776-f004:**
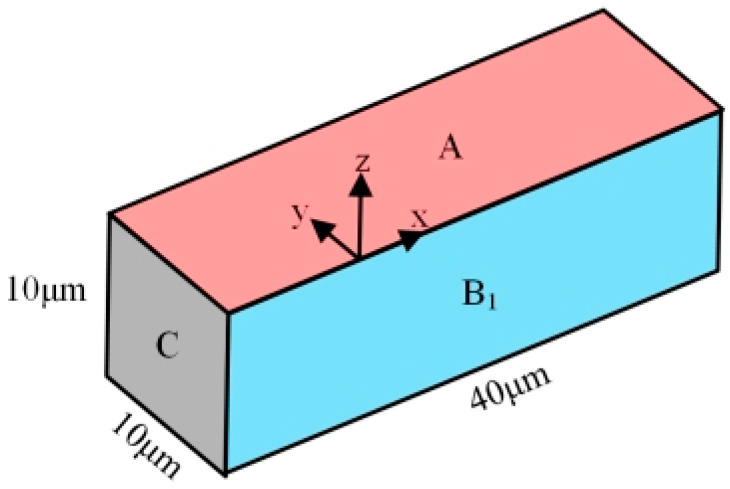
Simplified micro-element on the micro-bit surface.

**Figure 5 micromachines-13-00776-f005:**
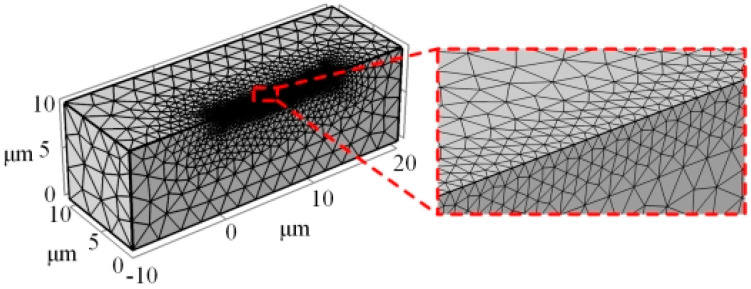
Finite element division of the micro-element of the micro-bit.

**Figure 6 micromachines-13-00776-f006:**
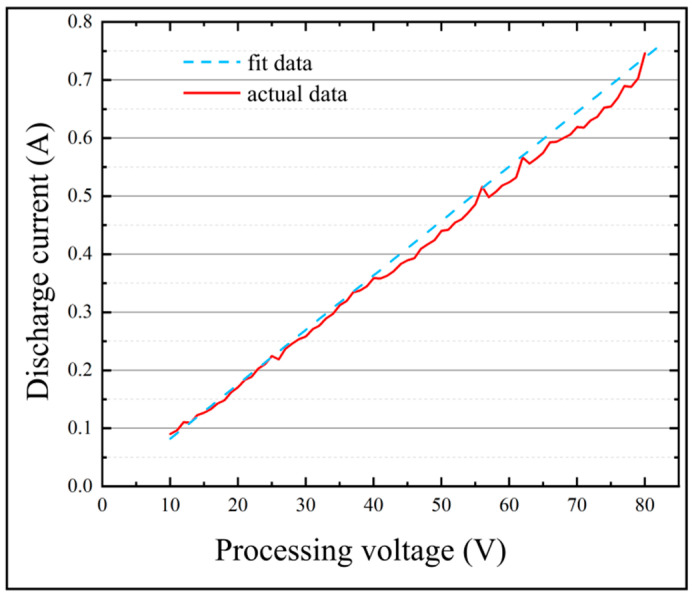
Results of correspondence between the discharge current and the machining voltage.

**Figure 7 micromachines-13-00776-f007:**
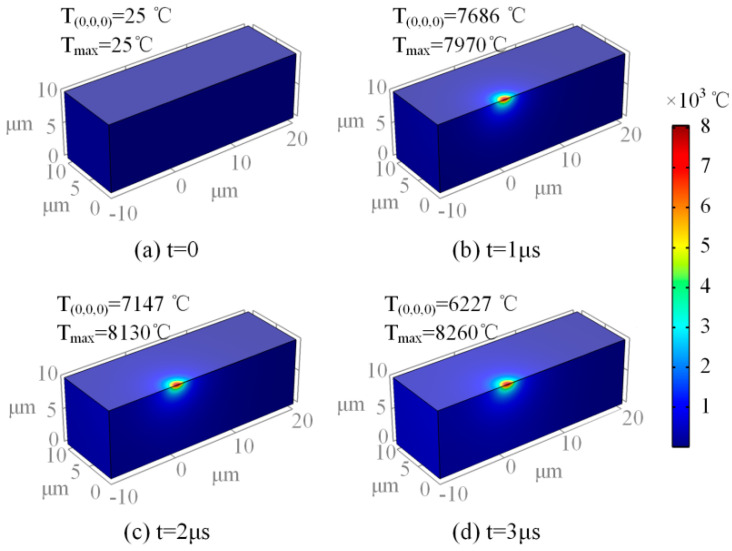
(**a**) Temperature field of micro-drill during discharge at t = 0, (**b**) temperature field of micro-drill during discharge at t = 1 μs, (**c**) temperature field of micro-drill during discharge at t = 2 μs, (**d**) temperature field of micro-drill during discharge at t = 3 μs.

**Figure 8 micromachines-13-00776-f008:**
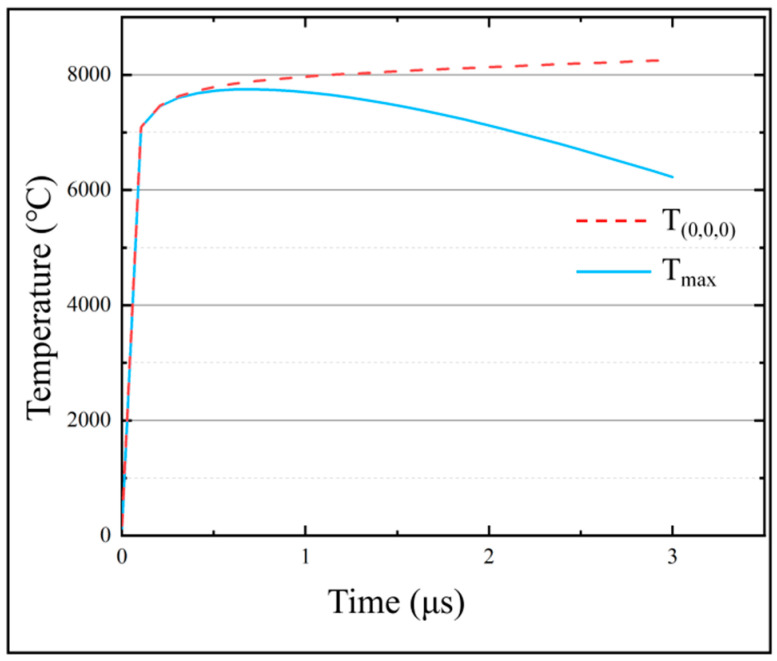
Variation of *T_max_* and *T*_(0,0,0)_ with machining time.

**Figure 9 micromachines-13-00776-f009:**
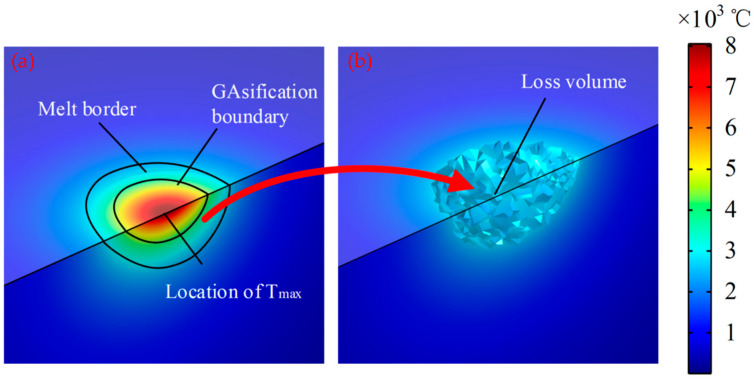
(**a**) Melting area of the micro-bit; (**b**) edm wear of the micro-bit.

**Figure 10 micromachines-13-00776-f010:**
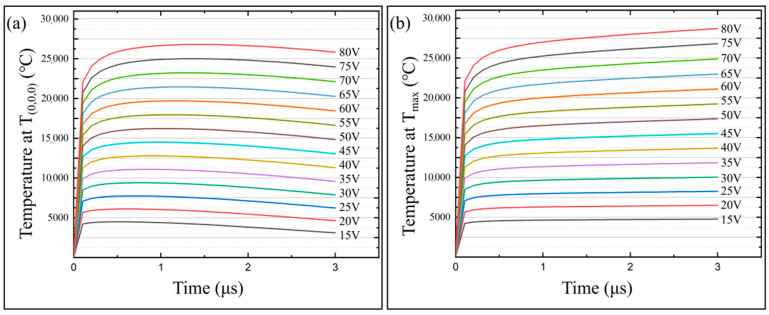
(**a**) *T*_(0,0,0)_ variation with machining time at different machining voltages; (**b**) *t_max_* variation with machining time at different machining voltages.

**Figure 11 micromachines-13-00776-f011:**
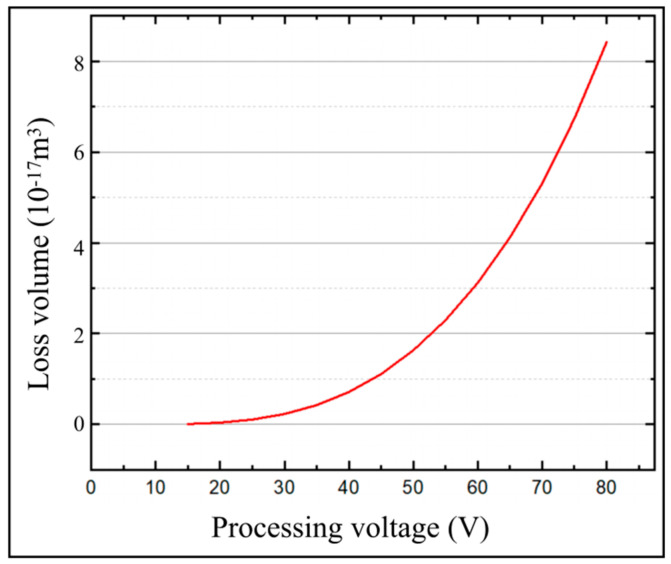
Micro-bit wear volume (*V_m_*) under different machining voltage.

**Figure 12 micromachines-13-00776-f012:**
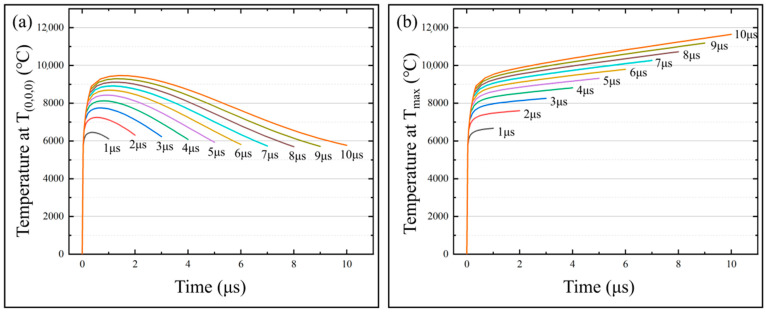
(**a**) *T*_(0,0,0)_ variation with machining time at different pulse widths; (**b**) *T_max_* variation with machining time at different pulse widths.

**Figure 13 micromachines-13-00776-f013:**
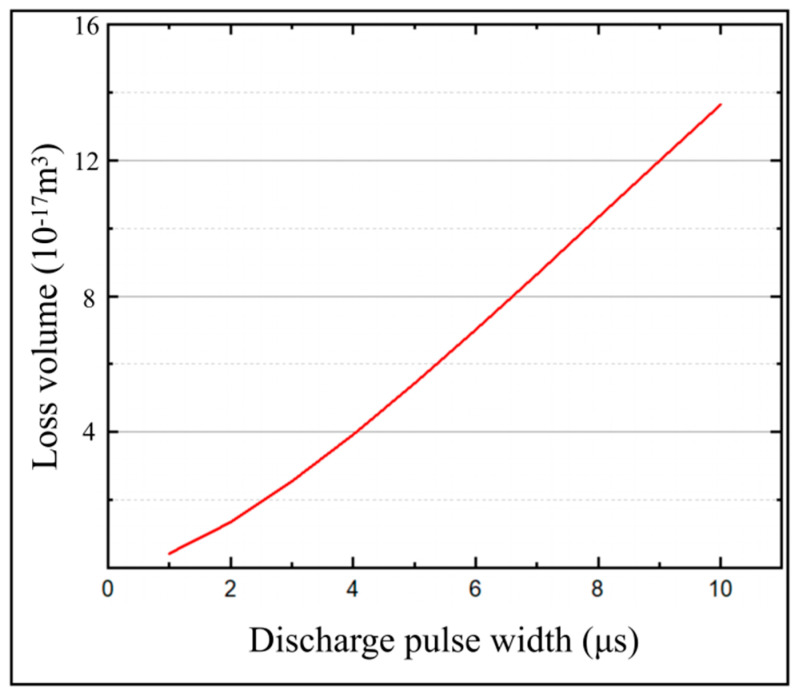
Micro-bit wear volume (*V_m_*) under different pulse width.

**Figure 14 micromachines-13-00776-f014:**
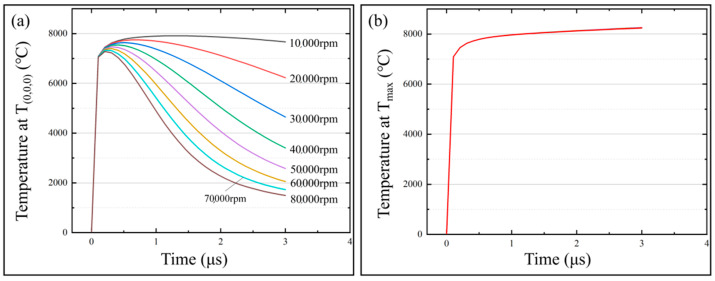
(**a**) *T*_(0,0,0)_ variation with machining time at different spindle speeds; (**b**) *T_max_* variation with machining time at different spindle speeds.

**Figure 15 micromachines-13-00776-f015:**
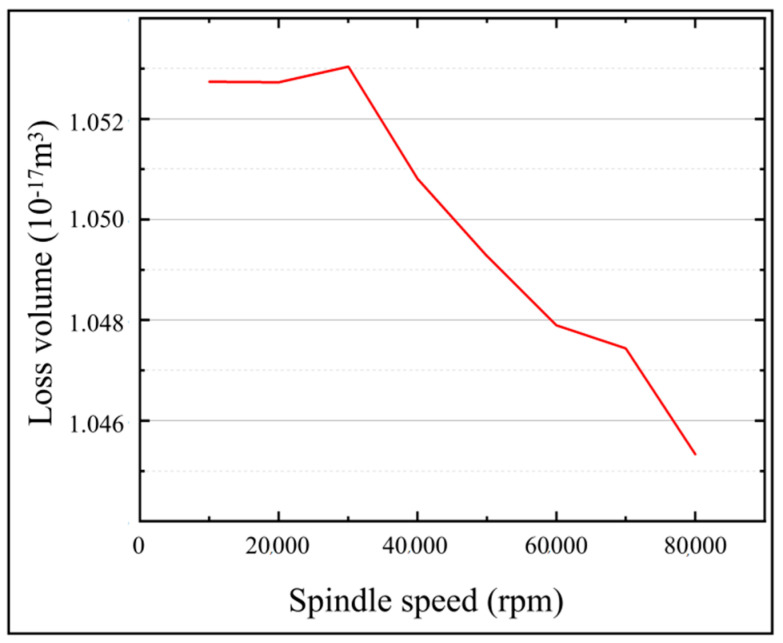
Micro-bit wear volume (*V_m_*) under different spindle speeds.

**Figure 16 micromachines-13-00776-f016:**
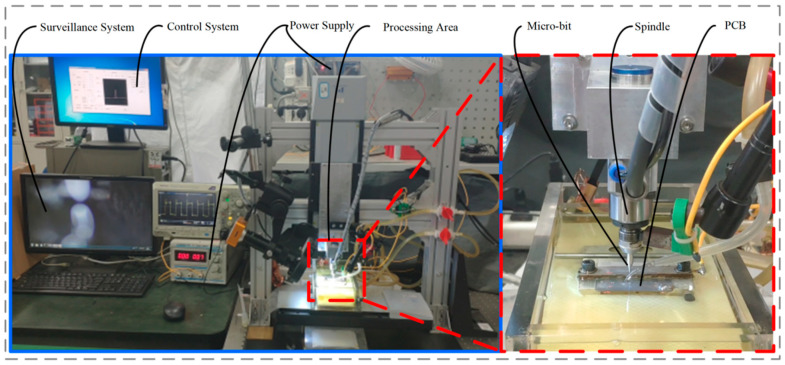
Laboratory Equipment.

**Figure 17 micromachines-13-00776-f017:**
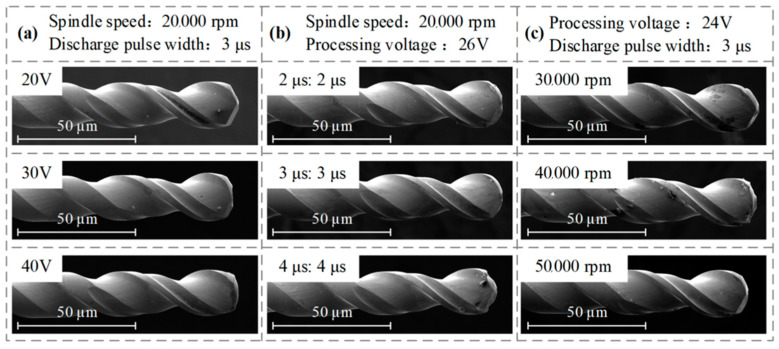
EDM wear of the micro-bit: (**a**) effects of machining voltages on the micro-bit wear; (**b**) effect of pulse widths on the micro-bit wear; (**c**) effect of spindle speeds on the micro-bit wear.

**Table 1 micromachines-13-00776-t001:** Material properties of the edge of the micro-bit.

Material Properties	Value
Density	14,500 kg/m^3^
Constant pressure heat capacity	209 J/(kg*K)
Thermal Conductivity	75.36 W/(m*K)
Melting point	2800 °C
Thermal expansion coefficient	4.5 × 10^−6^/°C
Hardness	89.5 HRA

**Table 2 micromachines-13-00776-t002:** Discharge current at different voltages.

Processing Voltage/V	Discharge Current/A	Processing Voltage/V	Discharge Current/A	Processing Voltage/V	Discharge Current/A	Processing Voltage/V	Discharge Current/A
10	0.090416	28	0.245740	46	0.393160	64	0.564680
11	0.096498	29	0.253440	47	0.409180	65	0.574560
12	0.110806	30	0.257980	48	0.417060	66	0.592720
13	0.109786	31	0.270880	49	0.424460	67	0.593780
14	0.122738	32	0.276700	50	0.440360	68	0.599980
15	0.126736	33	0.289440	51	0.442040	69	0.606080
16	0.133314	34	0.297040	52	0.454180	70	0.619160
17	0.142644	35	0.311840	53	0.460180	71	0.617940
18	0.148212	36	0.319100	54	0.472200	72	0.630120
19	0.162370	37	0.334100	55	0.486100	73	0.636820
20	0.170902	38	0.337540	56	0.516160	74	0.652580
21	0.183612	39	0.344600	57	0.497940	75	0.654200
22	0.188642	40	0.358600	58	0.507000	76	0.669060
23	0.202720	41	0.358280	59	0.518600	77	0.689520
24	0.210520	42	0.362780	60	0.523720	78	0.688400
25	0.224880	43	0.370780	61	0.531900	79	0.702660
26	0.218660	44	0.383220	62	0.566640	80	0.746360
27	0.236940	45	0.389480	63	0.556040	81	0.74931

## Data Availability

Not applicable.

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
