# Peer review of "Simulation of Temperature Field in Micro-EDM Assisted Machining of Micro-Holes in Printed Circuit Boards"

_micromachines, 2022, doi:10.3390/mi13050776_

Round 1

Reviewer 1 Report

The article is devoted to the actual problem of modern mechanical engineering.

The authors established the dependence of tool wear parameters.

Obtaining mathematical models that allow predicting the processing process.

Comments on the article:

  1. In the introduction, an analysis of the works of the authors in this area is carried out. The works of one author predominate. The competence of scientists in the reference is beyond doubt. But it is worth expanding the reference. Pay attention to the work of the authors Sidhu and Ablyaz.
  2. The presented equations 1-3 are basic in electrophysics. Why should they be indicated in the article and give their description. It is necessary to focus on the developed model by the authors.
  3. The Comsol software package allows you to simulate complex processes. To do this, this software package has various modules. What is the contribution of the authors to process modeling? You need to show the originality of your own approach. Put more emphasis on the algorithm and modeling technique.
  4. Equation 11 is a regression equation. This equation was obtained by experiment. How does this equation apply to the Comsol mathematical model ? The regression equation is adequate only in the range of experimental modes. Why did you select the range of process parameters?
  5. Figure 10 shows the temperature fields. These are theoretical calculations. At temperatures of 30,000 degrees, what happens in the breakdown channel? What phenomena occur? The authors simply describe their calculated data without describing the physics of the process.

Author Response

Dear Professor:

Thank you for your letter and for the reviewers’ comments concerning our manuscript entitled “Simulation of temperature field in micro-EDM assisted machining of micro-holes in printed circuit boards” (Manuscript ID: micromachines-1706038). These comments are all valuable and very helpful for revising and improving the paper. After checking and studying the comments carefully, we have made revisions in the manuscript. Revised portion are marked in red in the paper. The main corrections in the paper and the response to the reviewer’s comments are listed as the following.

Reviewers' comments:

Reviewer : The article is devoted to the actual problem of modern mechanical engineering. The authors established the dependence of tool wear parameters. Obtaining mathematical models that allow predicting the processing process.

(1) Comment: In the introduction, an analysis of the works of the authors in this area is carried out. The works of one author predominate. The competence of scientists in the reference is beyond doubt. But it is worth expanding the reference. Pay attention to the work of the authors Sidhu and Ablyaz.

Response: Thanks for your important and helpful comments. We have improved the introduction and added some new references to expand the current state of research in the this area. We also have paid attention on the work of the authors Sidhu and Ablyaz, such as the following article:

Ablyaz, T.R., et al., Analysis of Wire-Cut Electro Discharge Machining of Polymer Composite Materials, Micromachines (Basel), 2021, 12(5).

(2) Comment: The presented equations 1-3 are basic in electrophysics. Why should they be indicated in the article and give their description. It is necessary to focus on the developed model by the authors.

Response: Thanks for your comment. In this paper, the single discharge channel between the micro-bit and the copper was simplified as a cylindrical shape. The presented equations 1-3 were the basis for building the discharge model, so we have to indicate in the article and give their description. Equation 1 was used to calculate the total energy (Wt) generated by discharge per unit time. The energy generated by the discharge was distributed among the cathode, anode and process medium in a specific proportion, and Equation 2 was used to calculate the energy (We) distribute to the cathode. Equation 3 was used to calculate the average heat flux density (He) within the discharge channel on the cathode surface.

(3) Comment: The Comsol software package allows you to simulate complex processes. To do this, this software package has various modules. What is the contribution of the authors to process modeling? You need to show the originality of your own approach. Put more emphasis on the algorithm and modeling technique.

Response: Thanks for your comment. This paper established a heat conduction model in micro-EDM assisted machining of micro-holes in printed circuit boards, which was not part of the original module in the Comsol software. During the EDM process, the heat flux density of the discharge channel was not uniformly distributed, but conformed to be normal distribution. In this paper, the overall heat flux density distribution on the surface of the micro-bit was established through theoretical derivation and mathematical analysis, as shown in Figure 2.

(4) Comment: Equation 11 is a regression equation. This equation was obtained by experiment. How does this equation apply to the Comsol mathematical model ? The regression equation is adequate only in the range of experimental modes. Why did you select the range of process parameters?

Response: Thanks for your comment. Equation 10-11 were the results derived by other researchers, which were consistent with this experiment. In these equations, both the cathode energy distribution coefficient and the discharge channel radius are calculated with the current Id. However, due to the limitation of the equipment, the current Id can not be used as an independent variable during the experiment. Therefore, a series of discharge experiments were carried out using voltages ranging from 10V to 80V to determine the magnitude of the discharge current. After that, using the oscilloscope to measure the peak current in the discharge loop while the discharge state was stable. Finally the relationship between the voltage and the current can be obtained, as shown in Equation 12.

Before this research, a series of experiments have been conducted, and it was found that the quality of the micropores processed in the voltage range of 10-80V was relatively high, so the process parameters selected in this paper were required to be in range of 10-80V.

(5) Comment: Figure 10 shows the temperature fields. These are theoretical calculations. At temperatures of 30,000 degrees, what happens in the breakdown channel? What phenomena occur? The authors simply describe their calculated data without describing the physics of the process.

Response: Thanks for your comment. This paper established the simulation model by combining the method of experiment with simulation. Because the wear of the microdrill was composed of numerous discharge pits, the physical characteristics of the process were not described in the paper. This simulation was used to calculate the final microdrill loss results through Comsol software. In this paper, the material of the micro-drill was YG8 cemented carbide with a melting point of 2800℃. When the temperature generated by the discharge was higher than the melting point of the microdrill, it would be melted and vaporized. In order to study the wear of the micro-drill at the high temperature of the discharge, this chapter assumes that the micro-drill is considered to be lost when it was melted.

We tried our best to improve the English of the manuscript and made some changes in the manuscript. These changes will not influence the content and framework of the paper. We appreciate for Editors/Reviewers’ warm work earnestly and hope that the correction will meet with approval. Once again, thank you very much for your comments and suggestions.

Name:Kai Jiang

Sincerely yours,

Kai Jiang

Reviewer 2 Report

Dear author,

The paper "Simulation of temperature field in micro-EDM assisted machining of micro-holes in printed circuit boards" is very good and logical. And also, it is significant to study the micro-EDM was used as an auxiliary means for machining PCB micro-holes to effectively eliminate the defects such as hole burrs and nail head. This paper has been reviewed but it needs minor revision before accepted. The followings are the points need to modify.

  1. In abstract, it should have some results of the paper.
  2. In line 127, it should be 40*20*10μm. So does other place in the paper.
  3. In line 156, there is too many blank before the word “heat”.
  4. In Figure 7, (a)t=0, (b) t=1… should be like the caption of figure 10. So does other place in the paper.
  5. There should have the picture of EDM and process of the experiment.
  6. There should have a comma in every 3 digits. For example, 80000 rpm in line 284. It should be 80,000 rpm.
  7. This paper is a very good one. There should have more reference. Suggest that at least 20.

Author Response

Dear Professor:

Thank you for your letter and for the reviewers’ comments concerning our manuscript entitled “Simulation of temperature field in micro-EDM assisted machining of micro-holes in printed circuit boards” (Manuscript ID: micromachines-1706038). These comments are all valuable and very helpful for revising and improving the paper. After checking and studying the comments carefully, we have made revisions in the manuscript. Revised portion are marked in red in the paper. The main corrections in the paper and the response to the reviewer’s comments are listed as the following.

Reviewers' comments:

Reviewer #2: The paper "Simulation of temperature field in micro-EDM assisted machining of micro-holes in printed circuit boards" is very good and logical. And also, it is significant to study the micro-EDM was used as an auxiliary means for machining PCB micro-holes to effectively eliminate the defects such as hole burrs and nail head. This paper has been reviewed but it needs minor revision before accepted. The followings are the points need to modify.

(1) Comment: In abstract, it should have some results of the paper.

Response: Thanks for your comment. We have added some results in abstract, which was marked in red in the paper.

(2) Comment: In line 127, it should be 40*20*10 μm. So does other place in the paper.

Response: Thanks for your comment and I’m very sorry for this mistake. We have revised it in the paper and marked it in red.

(3) Comment: In line 156, there is too many blank before the word “heat”.

Response: Thanks for your comment and I’m very sorry for this mistake. We have revised it in the paper.

(4) Comment: In Figure 7, (a)t=0, (b) t=1… should be like the caption of figure 10. So does other place in the paper.

Response: Thanks for your important comment and suggestion. We have revised it in the paper.

(5) Comment: There should have the picture of EDM and process of the experiment.

Response: Thanks for your important comment and helpful suggestion. We have added the picture of EDM and process of the experiment as shown in figure 16.

(6) Comment: There should have a comma in every 3 digits. For example, 80000 rpm in line 284. It should be 80,000 rpm

Response: Thanks for your comment and I’m very sorry for this mistake. We have revised it in the paper and marked it in red.

(7) Comment: This paper is a very good one. There should have more reference. Suggest that at least 20.

Response: Thanks for your comment. We have added some recent references. They are listed as follows:

  • Tang, J., Yang, X., Simulation investigation of thermal phase transformation and residual stress in single pulse EDM of Ti–6Al–4V, Journal of Physics D: Applied Physics,2018, 51(13).
  • Yang, F., et al., Simulation and experimental analysis of alternating-current phenomenon in micro-EDM with a RC-type generator, Journal of Materials Processing Technology,2018, 255, 865-875.
  • Choubey, M., et al., Finite element modeling of material removal rate in micro-EDM process with and without ultrasonic vibration, Grey Systems: Theory and Application, 2020, 10(3), 311-319.
  • Ablyaz, T.R., et al., Analysis of Wire-Cut Electro Discharge Machining of Polymer Composite Materials, Micromachines (Basel), 2021, 12(5).
  • Li,Z., Tang J. and Bai J., A novel micro-EDM method to improve microhole machining performances using ultrasonic circular vibration (UCV) electrode. International Journal of Mechanical Sciences, 2020,175, 105574.

We tried our best to improve the English of the manuscript and made some changes in the manuscript. These changes will not influence the content and framework of the paper. We appreciate for Editors/Reviewers’ warm work earnestly and hope that the correction will meet with approval. Once again, thank you very much for your comments and suggestions.

Name:Kai Jiang

Sincerely yours,

Kai Jiang
